# The Role of Background Wind and Moisture in the Atmospheric Response to Oceanic Eddies During Winter in the Kuroshio Extension Region

**Yinglai Jia [1,2,\*], Longjing Chen [1,3], Qinyu Liu [1,2], Xiaohui Yang [1] and Yifei Wu [1]**

1    Physical Oceanography Laboratory, Ocean University of China, Qingdao 266100, China
2    Pilot National Laboratory for Marine Science and Technology (Qingdao), Qingdao 266100, China
3    Goldwind Science and Technology Co., Ltd., Beijing 100176, China
\*    Correspondence: jiayingl@ouc.edu.cn

**Abstract:** The role of background wind and moisture in the atmospheric response to oceanic eddies during winter in the Kuroshio Extension (KE) region is examined by numerical experiments (EXPs) using the Weather Research and Forecasting (WRF) model. We designed two sets of parallel experiments (dry and wet EXPs). The dry EXPs exclude the moisture in the air and the evaporation process. Each experiment differs only in the background wind speed during the initial condition. The wet EXPs include humidity in the initial condition and evaporation during the integration; the other settings are the same as the dry EXPs. The atmosphere in the two sets of EXPs are forced by the same mesoscale sea surface temperature anomaly which resembles the oceanic warm eddy in KE region. The results of these EXPs confirm that under weak background wind conditions, the atmospheric secondary circulation over oceanic eddies is driven by the pressure adjustment process due to weak advection. In the case of the dry run, the increase in background wind enhances the sea surface wind (SSW) by increasing vertical mixing. The convergence of SSW induces vertical motion and heating in the boundary layer, which further decreases the instability. The atmospheric secondary circulation in the dry run remains within the boundary layer. In wet EXPs, the atmospheric response is similar to that in dry runs when the background wind is very weak. When the background wind speed is increased to the climatology value (in KE region) or higher, the vertical motion triggers the precipitation process and diabatic heating above the boundary layer, and the heating in turn reinforces the upward flow.

**Keywords:** mesoscale oceanic eddy; atmospheric response; Kuroshio extension region; background wind

## 1. Introduction

The upstream Kuroshio Extension (KE) region is known as one of the most active eddy regions in the world. The ubiquitous eddy activity in this region produces significant sea surface temperature anomalies (SSTAs). In winter, the frontal meandering and eddy activities induce heat flux and transport moisture into the atmosphere. Increasing evidence has revealed that oceanic eddies can induce responses in both the marine atmospheric boundary layer (MABL) and the troposphere.

The atmospheric response to oceanic eddies is generally depicted as an in-phase sea surface wind (SSW) anomaly over eddy-induced SSTAs [1–5]. The heating effect of warm eddies causes local responses in the atmosphere, such as increases in turbulent heat flux, local ascending motion, and precipitation over warm eddies [4,6]. The change in SSW results in convergence/divergence in the two sides of the eddy and drives a vertical secondary circulation (SC) in the atmosphere ([7],

hereafter CJL17). The mechanism related to this type of response is known as the vertical momentum mixing (VMM) mechanism [8,9]. Based on a dry, two-dimensional configuration of the Weather Research and Forecasting (WRF) model, Kilpathrick et al.'s work revealed that the response of the atmosphere increases with the background wind speed because advection plays an important role in MABL dynamics [10]. In the case of 'weak' cross-front winds, the reduced gravity model results revealed a different mechanism in which the pressure gradient plays a larger role [2,11,12]. This type of atmospheric response is controlled by a mechanism known as the pressure adjustment mechanism (PAM), with is a typical sea level pressure (SLP) and vertical velocity response located over the eddy center [13]. Using idealized simulations of a dry atmosphere on the f-plane, Lambaerts et al. examined the MABL response to SST fronts [14]. The researchers found that under weak wind conditions, the vertical velocity in the MABL is linearly proportional to the SST Laplacian. This type of response is also found over KE meanders [15] and 10% of oceanic eddies in the KE region [7].

Using observation and reanalysis data, CJL17 provides observational evidence that oceanic eddies can induce atmospheric responses through both VMM and PAM. In their work, the researchers found approximately 10% of the total detected eddies in the winter KE region related to the PAM type of atmospheric response and 60% with the VMM type response. Based on idealized model experiments and budget analysis, the work done by Shan and Dong [16] further proved that the SLP adjustment mechanism is more important when the background wind is weak. Through a series of sensitivity experiments, they examined the role of background instability, wind speed, eddy size, and intensity in the atmospheric responses. Their work indicates that diabatic heating leads to strengthening of kinetic energy in MABL.

In addition to the MABL responses, the effect of oceanic eddies on the troposphere has also been detected in previous studies. Ma et al. [17,18] revealed that the vertical motion over eddies can reach up to 700 hPa annually on average, based on CFSR reanalysis. By determining the apparent heat source, CJL17 found that the upward moisture transport and corresponding latent heat release are responsible for the upward flow to reach higher than 500 hPa. Lambaerts et al. [14] demonstrated the importance of moisture and latent heat release in the growth of baroclinic instability. Ma et al. [17,18] also emphasized the importance of diabatic heating in the oceanic eddy-induced atmosphere–ocean interaction. These works suggest that both background wind and moisture are important for the eddy-induced atmospheric response.

To explain how the MABL response can be extended to a free atmosphere, the moisture process must be considered, and the effect of background wind in the situation of a wet atmosphere needs to be examined. However, in previous modeling work about atmospheric responses over eddies, seldom have examined the difference between atmospheric responses under dry and wet conditions. In our study, the roles of background wind, vertical moisture transport, and diabatic heating were investigated by conducting a series of idealized experiments using the WRF model. Two parallel sets of idealized experiments were conducted, each including three subexperiments, differing only in the strength of background wind. One set used a dry configuration (dry run), and the other included humidity in the initial condition and considered evaporation to investigate the role of moisture (wet run). First, we checked the evolution of atmospheric responses and then the effect of background wind on the vertical motion and instability. We also examined the combined effect of strong wind and diabatic heating on the eddy-driven SC. The rest of this paper is organized as follows: In Section 2, the model configuration and background atmosphere environment are briefly stated. In Section 3, the sensitivity of the atmosphere response to the background wind is described. In Section 4, the effect of moisture release and diabatic heating under different background wind conditions are illustrated. Finally, in Section 5, a summary and discussion are provided.

## 2. Model Configuration

A series of idealized experiments were conducted using the WRF model, which is a three-dimensional, fully compressible and nonhydrostatic mesoscale atmosphere model. The Advanced

Research WRF (ARW, version 3.7) [19] (p. 88) has been widely used for mesoscale air–sea interaction research. The experiments in this study included 1600 km (zonal) × 800 km (meridional) grids with a 20 km horizontal resolution and 65 sigma levels in vertical. As we focused on the atmospheric response over eddies (which have a typical horizontal scale of 100–500 km), the change in the Coriolis parameter with latitude was neglected. All cases were carried out on an f-plane geometry of a typical midlatitude f-parameter ($10^{-4}$ s$^{-1}$) with open lateral boundary conditions. The model configuration included the WRF Single-Moment 3-class (WSM3) microphysics scheme [20,21], Kain–Fritsch (KF) cumulus scheme [22], and the YSU planetary boundary layer scheme [23]. The Rapid Radiative Transfer Model (RRTM) and MM5 (Dudhia) scheme for longwave and shortwave radiation [24,25]. The land surface model was MM5 5-layer soil temperature model.

To verify the role of background wind and moisture in the atmospheric response to oceanic eddies, we first performed a set of dry-configured runs and then conducted the same sets of experiments (EXPs) with inclusion of moisture in the initial phase. For the dry run, the initial conditions (ICs) of these experiments were first constructed on the pressure levels, and then automatically interpolated into sigma levels by WRF [19]. There was only a uniform westerly on each given pressure level, and the zonal wind decreased linearly with pressure. Specifically, the zonal wind was as follows:

$$u(p) = u0 + (1000 - P)/25, \tag{1}$$

where u(p) is the zonal wind on the pressure level (p) and P is the pressure. The u0 in the equation is the zonal wind at 1000 hPa. Based on the observations provided by Chen et al. [7], we selected three typical wind speeds (u0), which were 0.5 m/s for the weak wind run (DU0), 8 m/s (approximately averaged surface wind speed in the winter KE), and 15 m/s for the stronger wind runs (DU8 and DU15) (Figure 1a). We summarize the design of the EXPs in Table 1. As shown in Equation (1), the three wind profiles differ only in the surface wind speed u0, and the vertical gradients of winds in the three EXPs are the same. The geopotential and density of the atmosphere were then computed based on the thermal-balanced Equations (2 and 3) with wind and the regional averaged geopotential in the winter KE region. The thermal balance relation is as follows:

$$\frac{\partial \Psi}{\partial y} = -fu, \tag{2}$$

$$\frac{\partial \Psi}{\partial p} = -\alpha, \tag{3}$$

where $\Psi$ is the geopotential height (Figure 1b), $f$ is the Coriolis parameter, $u$ is the zonal wind, and $\alpha$ is the specific volume. $\Psi$ can be calculated from Equation (2) based on the wind profile. Then, we obtained $\alpha$ from Equation (3). With $\alpha$ and $p$, the potential temperature field was then calculated using the state Equation:

$$p = p_0 \left(\frac{R_d \theta}{p_0 \alpha}\right)^{\gamma}, \tag{4}$$

where $\theta$ is the potential temperature, $R_d$ is the dry air constant (287 J K$^{-1}$ kg$^{-1}$), $\gamma = 1.4$ and $p_0 = 1000$ hPa. Because there is no meridional wind in the ICs, the thermal fields are uniform in the zonal direction.

**Table 1.** Design of dry and wet experiments (EXPs).

| Dry EXPs | Wet EXPs | Background Wind Speed (m/s) |
|---|---|---|
| DU0 | WU0 | 0.5 |
| DU8 | WU8 | 8 |
| DU15 | WU15 | 15 |

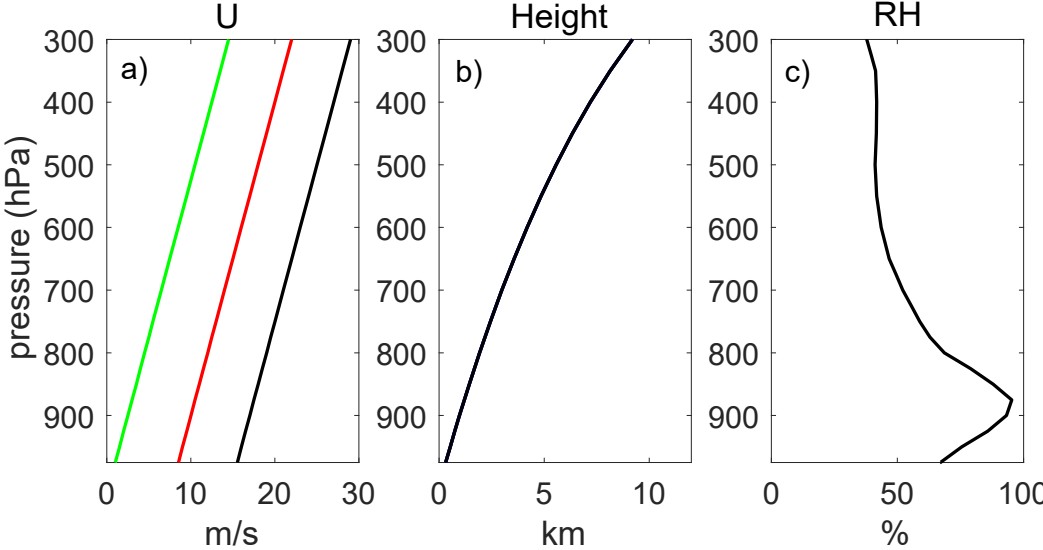

**Figure 1.** Vertical profiles of (**a**) zonal wind designed for the initial condition of the three EXPs, (**b**) spatially averaged geopotential height during winter in the Kuroshio Extension (KE) region, and (**c**) vertical profile of relative humidity (RH) in the wet air experiment.

Notably, there is no moisture in the dry-configured ICs, i.e., the relative humidity (RH) is set to zero. The evaporation at the lower boundary is also turned off. Therefore, the dry run EXPs exclude the influence of moisture and corresponding latent heat, which would allow us to focus on the role of background wind. In addition to the dry run, we also carried out wet run EXPs. In the IC of the wet run, the RH is uniform at each pressure level and varies with altitude. The RH profile is obtained from the average RH in the winter KE (Figure 1c). The water vapor mixing ratio is then calculated based on the moist state equation. The evaporation is also turned on in the wet run EXPs.

After constructing the ICs, we added a Gaussian-shaped warm bubble SST anomaly (SSTA) at the center of the grid. The distribution of warm bubbles is as follows:

$$SSTA = T_0 \cdot \exp\left(-\frac{r^2}{R^2}\right), \tag{5}$$

where $T_0$ is the amplitude of the SSTA, $R$ is the radius, and $r$ is the distance away from the center of the grid. The amplitude of $T_0$ is set to 3 K, and the radius $R$ is 100 km, which is the typical value of the eddy radius in the KE region [5,7]. This warm bubble is nearly the strongest compared to the observation that Chen et al. [7] reported, but is still a reasonable assumption. Note that the atmosphere is also sensitive to the SST gradient, and the maximum SST gradient is located at approximately 70 km, with a magnitude of 3 K km$^{-1}$. This SST gradient is comparable to that of the SST fronts and allows our result to be as robust as possible.

## 3. Effect of Background Wind

To investigate the evolution of atmospheric responses in different background wind situations, we first checked those in the dry run. Within 1 h of simulation, the patterns in the three EXPs are similar, where all the responses, including SSW, SLP, planetary boundary layer height (PBLH) and vertical velocity, are located above the center of the eddy (Figure 2). The amplitudes of these responses increase with the background wind. Compared with the 10-h output (Figure 3), the amplitudes of SSW and PBLH (related to vertical mixing) anomalies are approximately half as large, while the amplitudes of vertical speed and SLP are less than 1/5 of those in the 10-h simulation. This suggests that the vertical mixing mechanism responds more quickly than the advection and thermal mechanisms. In the

initial stage of simulation, the effect of background wind is to induce vertical mixing and downward momentum transport.

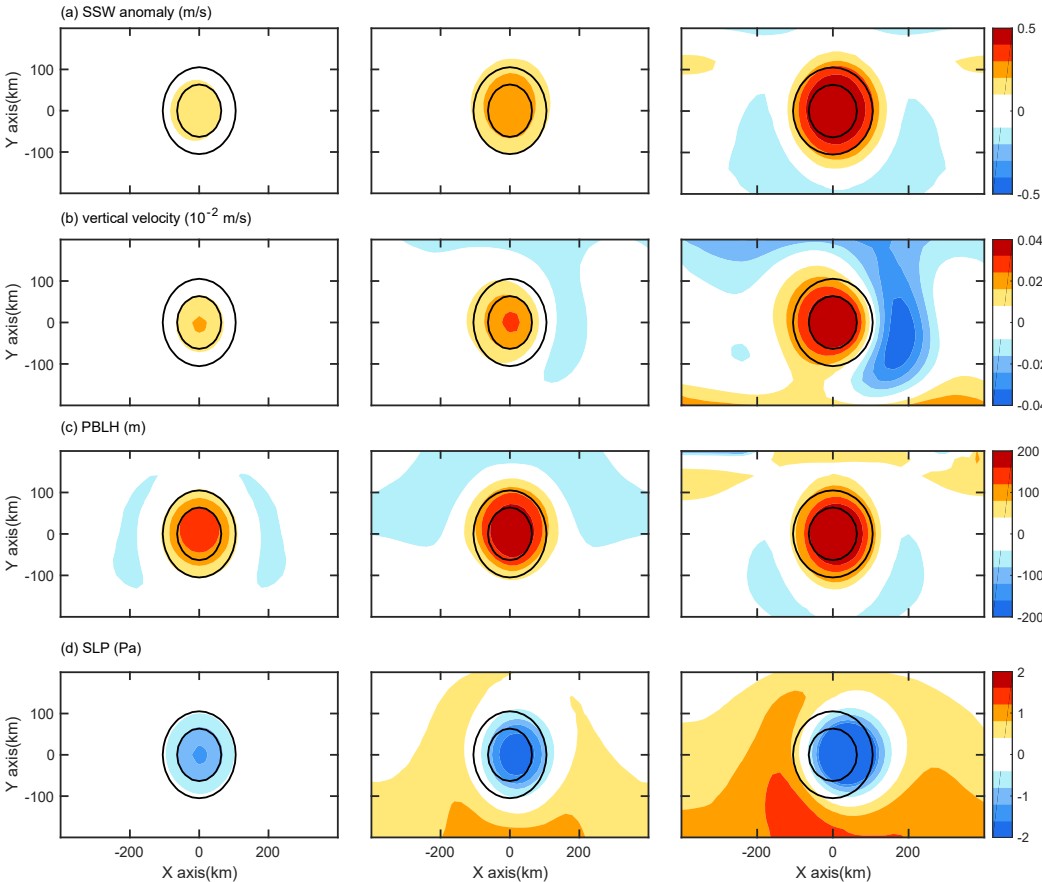

**Figure 2.** (**a**) Sea surface wind (SSW) speed anomaly, (**b**) vertical velocity on 900 hPa, (**c**) planetary boundary layer height (PBLH), and (**d**) sea level pressure (SLP) from the 1 h output in the three dry run experiments. The left column is for DU0, the middle column is for DU8, and the right column is for DU15. The units are as follows: m s$^{-1}$ for wind speed, $10^{-2}$ m/s for w, Pa for SLP, and m for PBLH. Contours in (**a–d**) are the 1 °C (outer contour) and 2 °C (inner contour) isolines of the sea surface temperature (SST) anomaly, which indicate the location and shape of the oceanic eddy.

The atmospheric response pattern between the weak wind runs and stronger wind runs began to differ from each other when the simulation proceeded. Figure 3 shows the 10-h responses in the three dry EXPs. The position of the responses in the weak wind run (DU0) differs from those in DU8 and DU15: The SSW anomaly shifted to the upwind side, while the vertical velocity and SLP anomalies remained in the eddy center. This resembles the PAM type of response. In the EXPs with stronger wind, the in-phase distribution of SSW and downwind shifted vertical velocity anomalies indicate the dominance of VMM. This agrees with previous works [11,12,16], where the weak wind is the favorite condition for PAM.

The SSW response in DU15 is approximately two times larger than those in DU0 and DU8. As in our EXPs, the vertical wind shear in the initial condition does not change with the increase in wind speed; thus, the significant increase in SSW should not be induced by the vertical shear instability of the initial condition. As indicated in the work by Shan and Dong [16], this is related to the increased vertical transport of turbulent kinetic energy (TKE) with the increase in background wind. The largest PBLH anomaly in DU15 proved that the vertical mixing increases with the background wind. The PBLH response remains within the eddy center in the three EXPs, suggesting that the vertical mixing is anchored above the SSTA due to the heating effect. This agrees with the work by Shan and Dong [16],

where the TKE response and vertical mixing are also significant at the eddy center. The SLP anomaly shifted to the downwind side in DU8 and DU15, with the amplitude remaining similar in the three EXPs. The SLP amplitude did not change with the increase in wind speed because the SLP is sensitive to the SSTA value, not the background wind [16]. However, what is the mechanism of the downwind shift in SLP in DU8 and DU15?

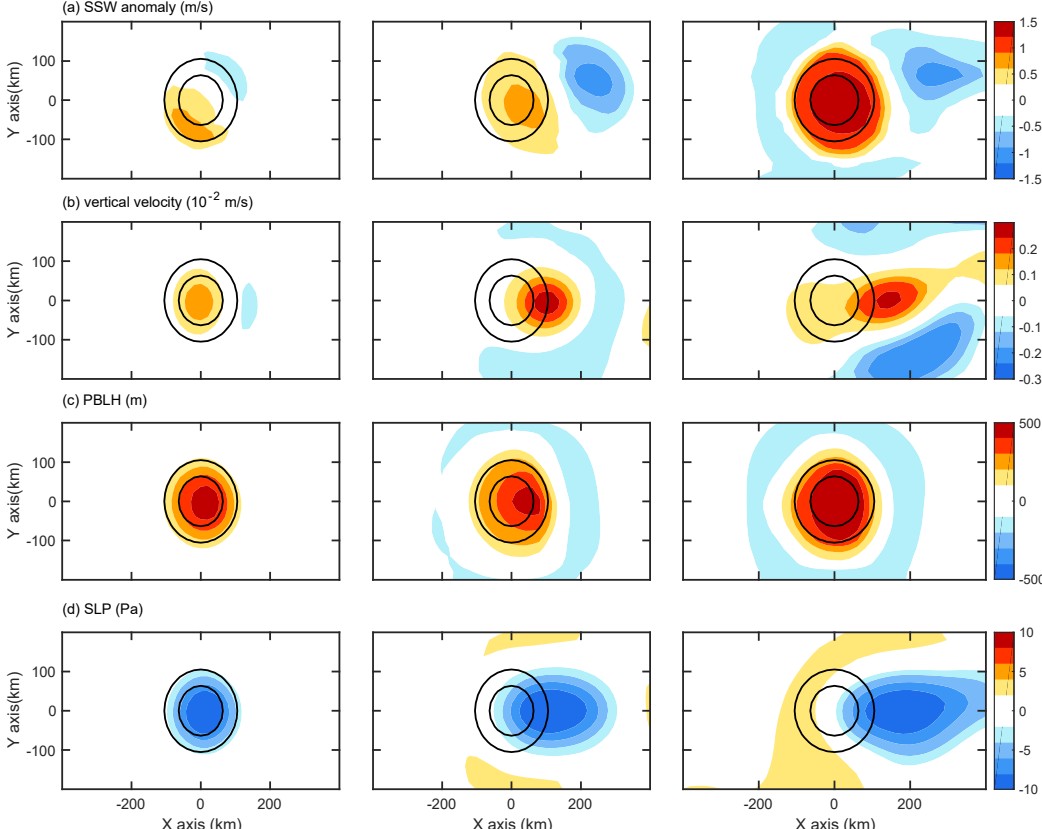

**Figure 3.** (**a**) SSW speed anomaly, (**b**) vertical velocity on 900 hPa, (**c**) PBLH, and (**d**) SLP from the 10-h output in the three dry run experiments. The left column is for DU0, the middle column is for DU8, and the right column is for DU15. The units are as follows: m s$^{-1}$ for wind speed, $10^{-2}$ m/s for w, Pa for SLP, and m for PBLH. Contours in (**a**–**d**) are the 1 °C (outer contour) and 2 °C (inner contour) isolines of the SST anomaly, which indicate the location and shape of the oceanic eddy.

To answer this question, we checked the evolution of potential temperature (PT), vertical motion and SLP anomalies (Figure 4) in DU0 and DU15, and then compared the advection terms in the equation of potential temperature in the two dry EXPs (which is the apparent heat source equation in Yanai et al. [26]; Yanai and Tomita [27]; and Kobashi et al. [28]). The equation is listed below:

$$Q_1 = C_p \left(\frac{p}{p_0}\right)^{\frac{R}{C_p}} \left(\underbrace{\frac{\partial\theta}{\partial t}}_{(1)} + \underbrace{V\cdot\nabla\theta}_{(2)} + \underbrace{\omega\frac{\partial\theta}{\partial p}}_{(3)}\right) \tag{6}$$

where $\theta$ is the PT, $V$ is the horizontal velocity, $\omega$ is the pressure vertical velocity, $p$ is the pressure, $p_0 = 1000$ hPa, $R$ is the gas constant, and $C_p$ is the specific heat capacity of dry air.

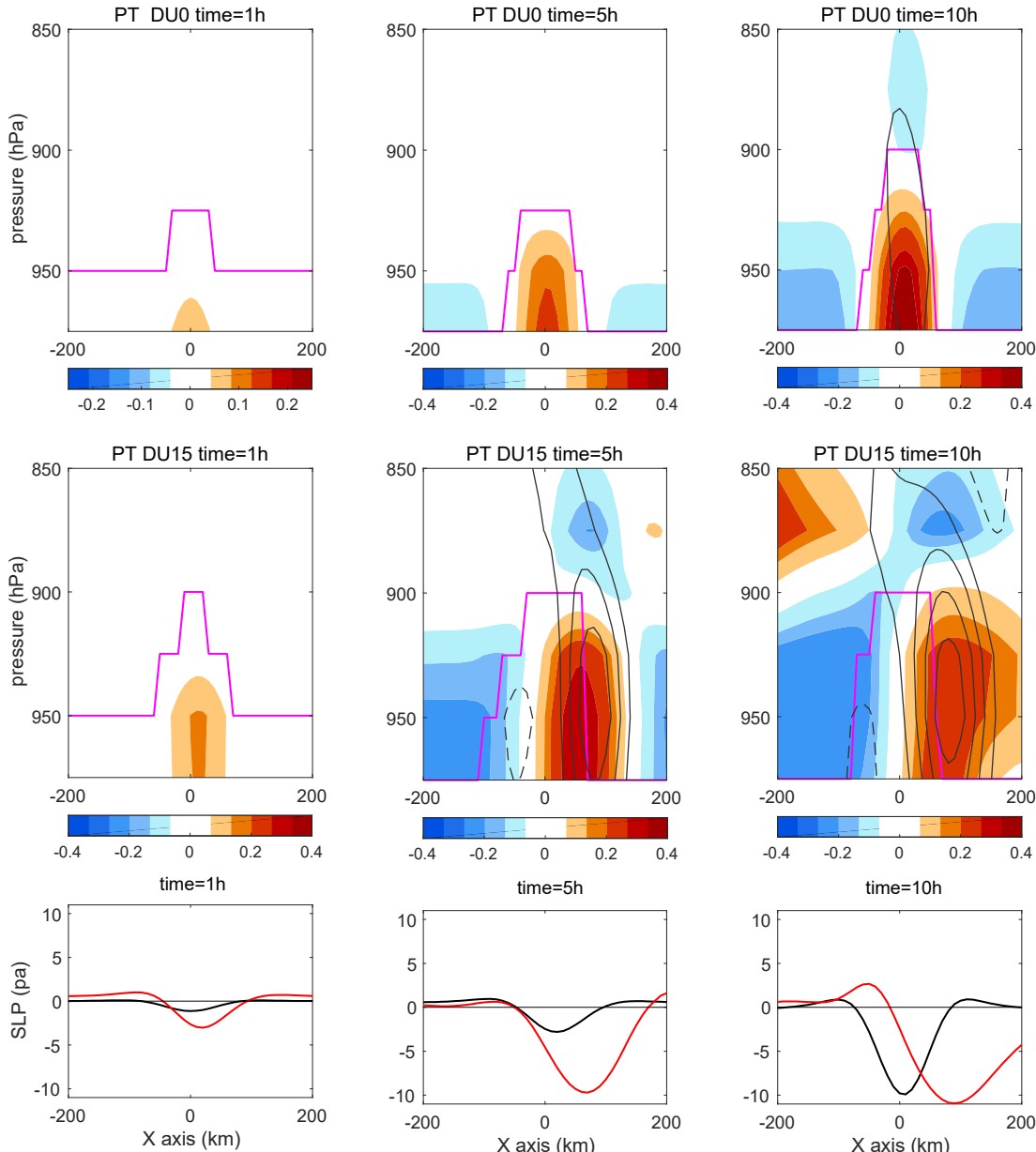

**Figure 4.** Time evolution of potential temperature (PT) (color, unit: K) anomalies and vertical velocity W (contour with 0.2 interval, unit: $10^{-2}$ m/s) in DU0 (top), DU15 (middle), and SLP anomalies (the bottom row, black lines are for DU0 and red lines for DU15, unit: pa). Left column: output of the 1-h output, middle column: 5-h output, and right column: 10-h output. Magenta line marks the position of PBLH.

The atmospheric responses in DU0 (including the vertical velocity, PT, and SLP anomalies) all remain within the eddy region, while they shifted to the downwind side in DU15 (Figure 4). In the weak wind case, the horizontal (term 2) and vertical advection of heat (term 3) are weak, and the increase in PT is mainly due to the local heating forced by the SST anomaly in the eddy center (Figure 5). In the strong wind case, in DU15 for example, during the early stage of simulation (in the first hour), the PT anomaly appears above the eddy center, as in the weak wind case, because of the heating effect of the SST anomaly. However, as the integration proceeded, the advection term became more important. Although the heating source still lies above the eddy center, the horizontal heat advection increased the PT at the downwind side and caused a shift in the PT anomaly (Figures 4 and 5). Thus, for dry atmospheric heating, SSTA is important when advection is weak. With the increase in background

wind speed, the advection effect becomes more important. Thus, the SLP anomaly shifted downwind in stronger wind EXPs.

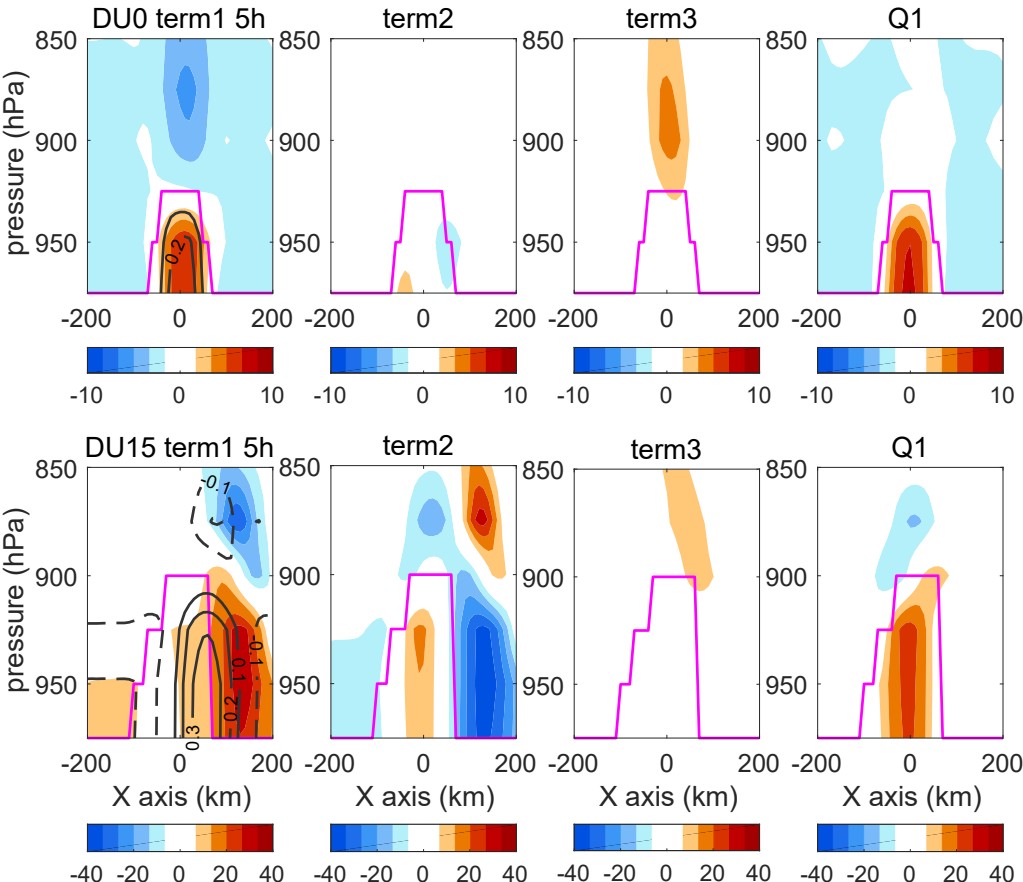

**Figure 5.** Terms in the Q1 Equation (Equation (6), color, multiplied by $10^{-3}$) and PT anomaly (black contours in the first figure in each row, unit: K) for the 5-h simulation in DU0 (top) and DU15 (bottom). Magenta line marks the position of PBLH. The time derivative in term1 is computed on every 10 min.

What is the effect of the increased PT on the downwind side? To answer this question, we calculated the Eady growth rate maximum according to Equation (7) (indicating baroclinic instability, Shaman et al. [29]). In Equation (7), $f$ is the Coriolis parameter, $u$ is the zonal wind velocity, and $N$ is the Brunt–Väisälä frequency. $\frac{\partial u}{\partial z}$ is the vertical shear of zonal wind between each two adjacent levels. According to the classic baroclinic theory, baroclinic instability can be measured by the Eady growth rate. As shown in Equation (7), the Eady growth rate is related to both vertical wind shear and $N^{-1}$. Because $N^{-1}$ is much larger (about 1000 s$^{-1}$) than the vertical shear of $u$ (about $10^{-3}$ s$^{-1}$), we examined the $N^{-1}$ and Eady growth rate to evaluate the role of PT variations in the growth of vertical motion over eddies.

$$\sigma_{BI} = 0.31f\left|\frac{\partial u}{\partial z}\right|N^{-1} \tag{7}$$

Figure 6 shows the increase in baroclinic instability in the 10th hour with the increase in background wind. Among the three EXPs, the vertical motion in DU0 is the weakest, and the vertical range of the flow mainly remains in PBLH. In DU0, the Eady growth rate is mainly distributed in the eddy region and spread under the PBL. In stronger wind cases (DU8 and DU15), the Eady growth rate increased by 2–3 times and the maximum shifted to the downwind side, corresponding with the upward flow. This suggests the effect of baroclinic instability on vertical motion in stronger wind cases. A large $N^{-1}$ value was also distributed at the eddy center in DU0 when advection was weak. However, in DU8 and DU15, a larger value of $N^{-1}$ was found on the downwind side over eddy. These results show that the

increase in background wind transports the heat to the downwind side, increases PT and decreases instability, and then increases vertical motion. In turn, the increased vertical motion reinforces the PT and reduces instability. These dynamic and thermal effects are additive, as suggested by Song et al. [30]. However, in dry run EXPs, the response of the atmosphere mainly remains within the PBLH. To induce upward motion in the free troposphere, additional heating above PBL is necessary. What is the wind effect when the moisture flux and diabatic heating are included?

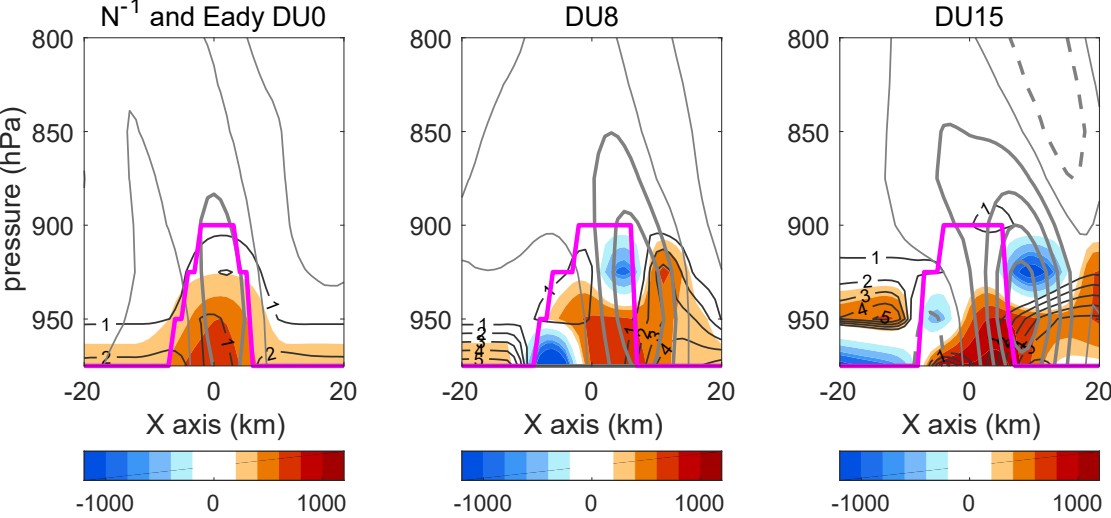

**Figure 6.** Vertical profile of $N^{-1}$ (color, unit: $s^{-1}$), Eady growth rate (black contour, unit: $day^{-1}$), and vertical velocity (gray thick lines, unit: $10^{-2}$ m/s, contour interval: 0.2; thin gray lines are the zero line) from the 10-h output in the dry experiments DU0, DU8, and DU15. The magenta line marks the position of PBLH.

## 4. Effects of Moisture and Diabatic Heating

### 4.1. Comparison with the Dry Run

To investigate the role of water vapor phase change and diabatic heating in the eddy-induced vertical circulation, another set of EXPs (wet run) was conducted. The initial temperature and geopotential height were the same as those in the dry run EXPs. As the integration proceeded, the only difference between the dry and wet runs was due to the diabatic heating and moisture processes.

Figure 7 shows the 10th hour responses in the wet run EXPs. Compared with those in the dry run, the most significant change was the increase in vertical motion and SLP anomalies (both increased in value by more than 2 times). In contrast, the SSW anomaly decreased to 1/3 compared with those in the dry run, while the PBLH anomaly even decreased to a negative value on the downwind side of the eddy. Furthermore, SSW did not increase with increasing background wind speed as it did in the dry run. This suggests the suppression of vertical mixing in stronger wind cases (WU8 and WU15). We discuss this further in Section 4.3. In the wet run, the position of atmospheric responses is similar to those in the dry run. This indicates that the position of the response is regulated by the SSTA position. The pattern of atmospheric responses also did not change by including diabatic heating. In the wet run, PAM remained significant in the weak wind case and VMM was parallel in the stronger wind cases. What is the role of diabatic heating in the atmospheric response to eddies?

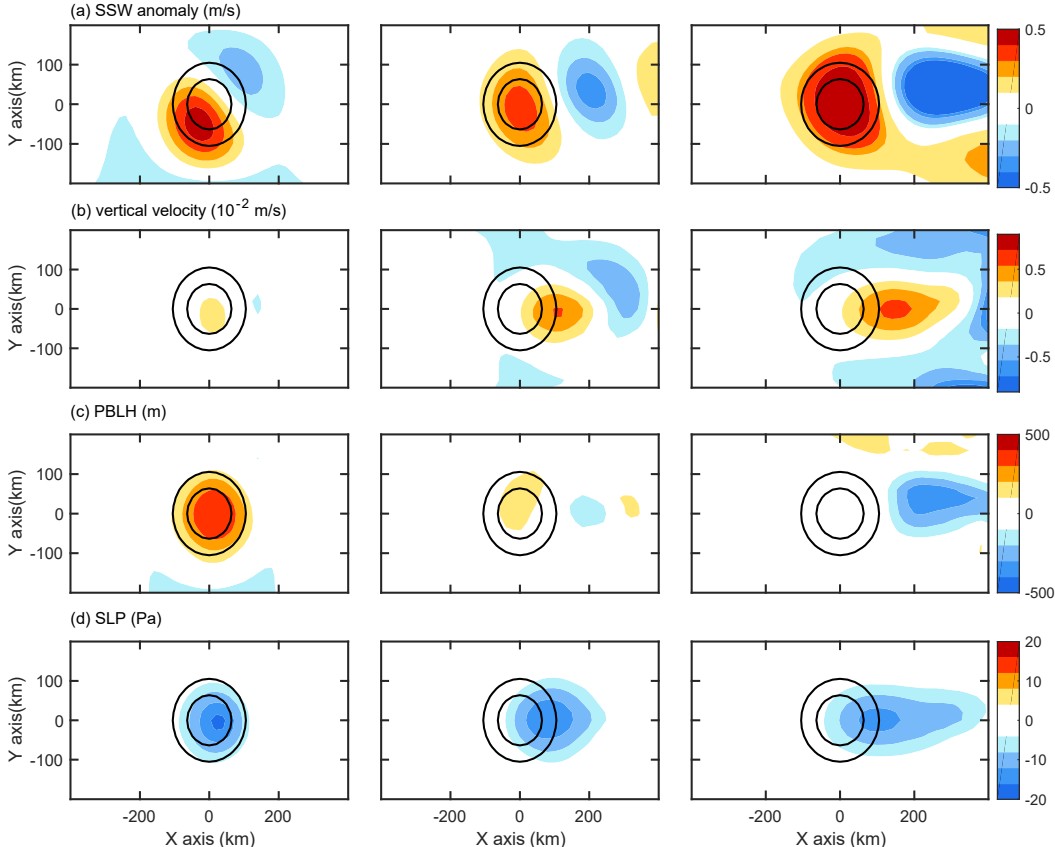

**Figure 7.** (**a**) SSW speed anomaly, (**b**) vertical velocity on 900 hPa, (**c**) PBLH, and (**d**) SLP from the 10-h output in the three wet run experiment 10-h outputs. The left column is for WU0, the middle column is for WU8, and the right column is for WU15. The units are as follows: m s$^{-1}$ for wind speed, $10^{-2}$ m/s for W, Pa for SLP, and m for PBLH. Contours in (**a**–**d**) are the 1 °C (outer contour) and 2 °C (inner contour) isolines of the SST anomaly, which indicate the location and shape of the oceanic eddy.

By comparing the vertical velocities in the wet run and dry run (Figures 6 and 8), it is clear that in the wet run, the upward flow grows much higher than in the dry run. This increase in SC is much more significant in stronger wind cases. The vertical circulation can reach up to 800 hPa in WU8 and WU15, which is high above the boundary layer (900 hPa). Along with the upward vertical motion, the cloud raises and diabatic heating (H) occurrs, especially in WU15 (Figure 8). Interestingly, in the stronger wind cases, the downward branch of the SC is also much stronger than those in the dry run, which makes the SC in the wet run look more distinct. We also checked the dry baroclinic instability indicator (Eady growth rate) in the wet run. Similar to the suppressed PBLH anomaly, the Eady growth rate decreased to less than half compared with the dry run. This suggests that mechanisms other than dry baroclinic instability play a role in driving the SC in the wet run. Then, how did the moisture processes affect the SC above the eddies?

### 4.2. Vertical Transport of Moisture

Recently, the eddy's effect on surface latent heat flux has been revealed by an increasing number of researchers [5–7]. However, the eddy's effect on the vertical transport of moisture, which is important to precipitation and tropospheric diabatic heating, has not been fully understood. In this section, we examine the vertical moisture flux and diabatic heating in a wet run. We calculated w' (perturbed vertical velocity) and q' (perturbed water vapor mixing ratio) by removing the zonal mean; then, the vertical moisture flux was calculated by <w'q'> in each EXP. From Figure 9, the vertical moisture flux occurred earlier in stronger wind cases. In the weak wind case, <w'q'> appeared after the 5th

hour of simulation and remained under 900 hPa within the boundary layer. In WU8 and WU15, this flux term appeared in the 2nd hour of simulation and increased with time as it extended to a level above 800 hPa. The rising center of vertical moisture flux suggests the upward transport of moisture over time.

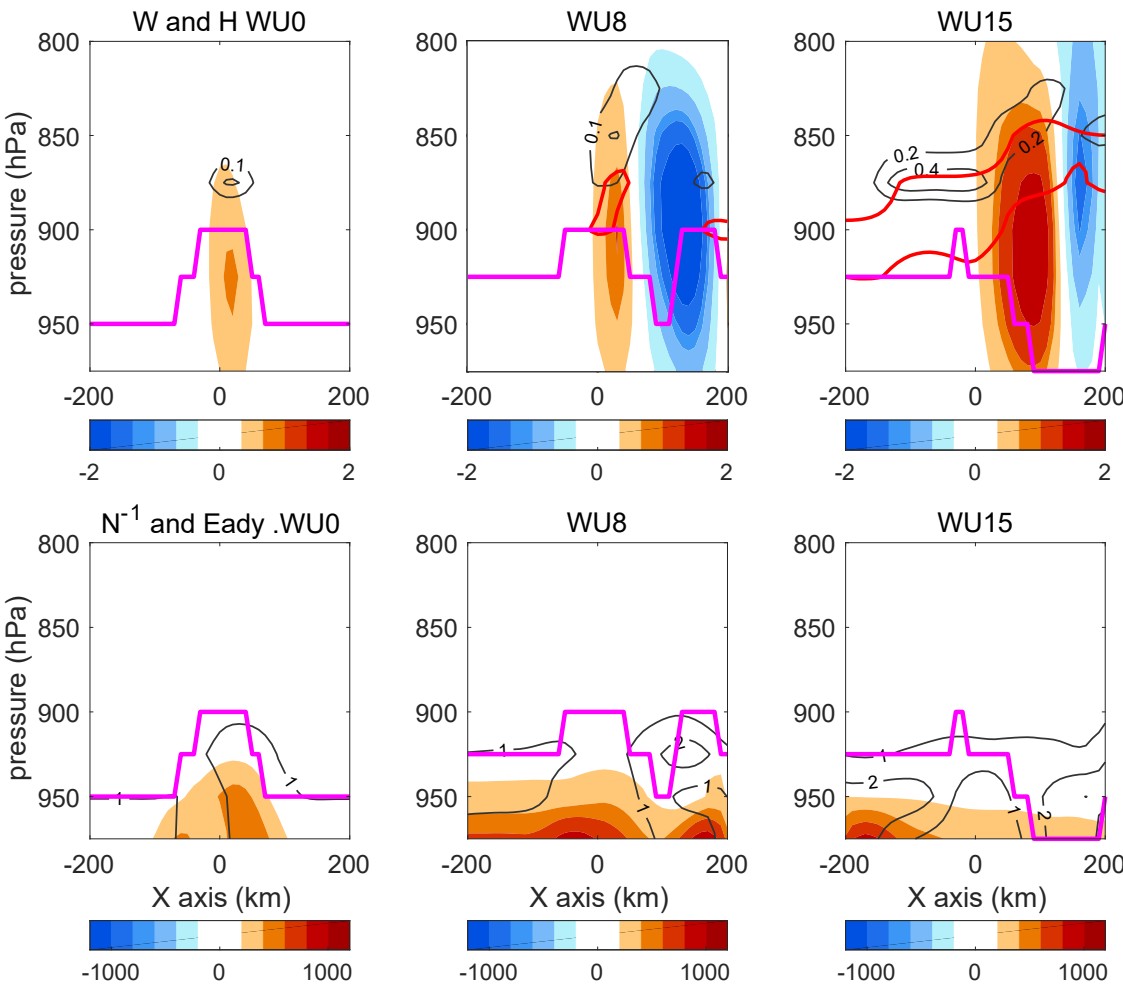

**Figure 8.** Top: Vertical profile of vertical velocity W (color, unit: $10^{-2}$ m/s), diabatic heating (H) (contour, unit: $10^{-3}$ K/s), and cloud fraction (red line marks the fraction $\geq$80%); bottom: $N^{-1}$ (color, unit: s) and Eady growth rate maximum (contour, unit: day$^{-1}$) in the wet run 10-h output. The magenta line marks the location of PBLH.

As the moisture flux increased with the background wind speed, we surmise that the stronger vertical mixing in WU15 introduced a positive feedback between the upward flow and diabatic heating due to the water vapor phase change in the troposphere. Without vertical motion, H is very weak in the weak wind case. However, it increased to a higher level in stronger wind cases. The vertical velocity in stronger wind cases (Figure 10) also increased with the background wind. From Figures 9 and 10, the vertical position of positive H agrees well with that of the moisture flux and the upper range of W. It is also interesting that the rapid increase in H occurred approximately 2 h later than that of the moisture flux, but occurred together with the precipitation. As the vertical motion increased faster after the onset of diabatic heating, we hypothesize that a positive feedback is triggered between vertical motion and diabatic heating. Furthermore, this positive feedback is strongest in WU15 when the background wind is the strongest.

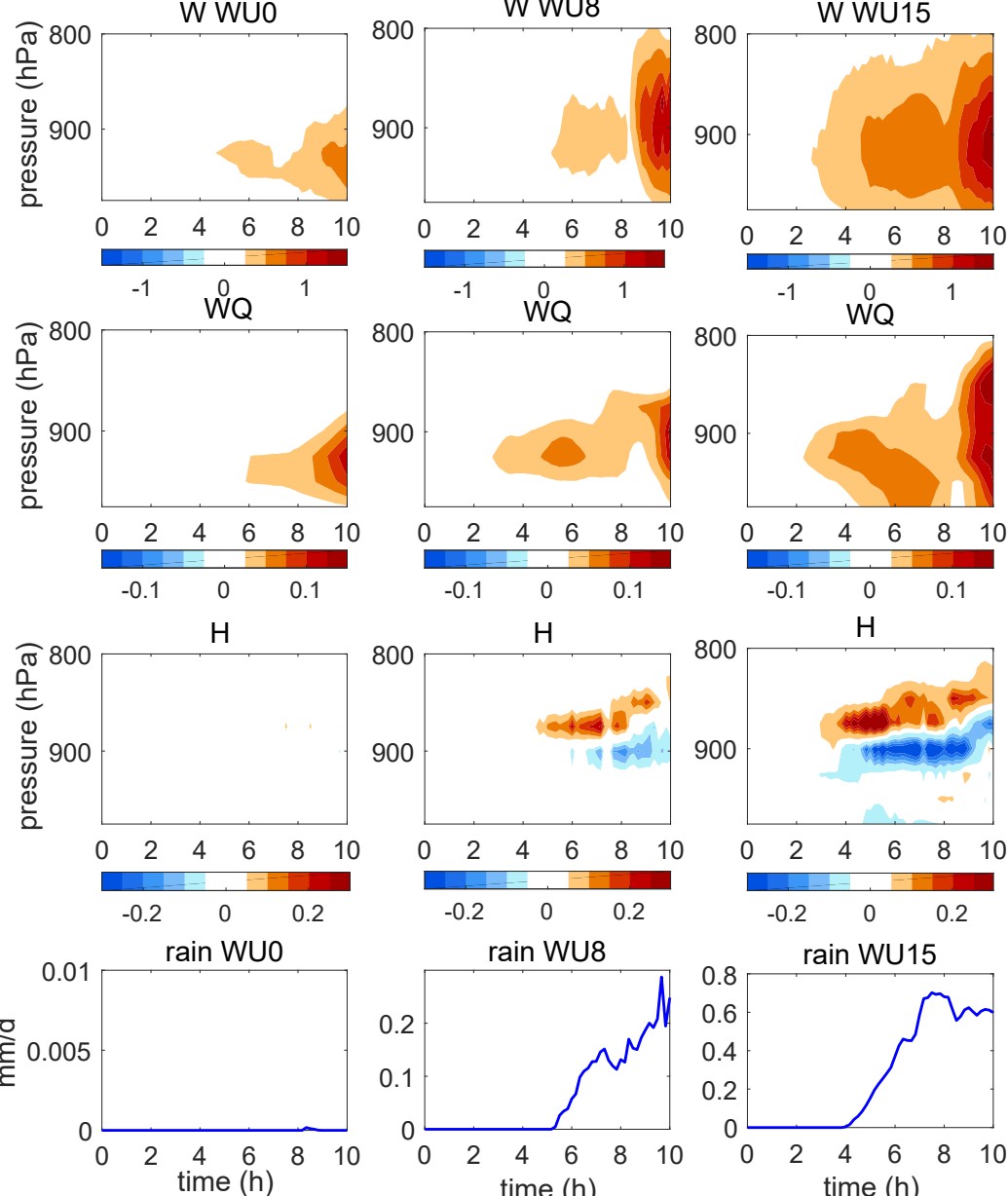

**Figure 9.** First row: Time evolution of vertical velocity (W) in wet runs. Second row: Time evolution of vertical transport of moisture (w'q') in wet runs. Third row: Time evolution of diabatic heating (H). Fourth row: Time evolution of rain rate. W, w'q', and H are averaged values from the eddy center to the 150 km downwind side along y = 0 km. The rain rate is averaged from the eddy center to the 150 km downwind side along y = 0. Left column: Results of experiment WU0; middle column: Results of experiment WU8; and right column: Results of experiment WU15. Units: $10^{-2}$ m/s for W, $10^{-2}$ g/kg m/s for w'q', $10^{-3}$ K/s for H, and mm/d for rain rate.

In the weak wind case, as there was almost no rain and H occurred, the response of the wet run resembles that of the dry run. In stronger wind cases, positive feedback was triggered, and strong vertical circulation was induced. This proves the importance of the background wind in the onset process of diabatic heating. The role of the background wind in the wet run is to induce strong upward flow by SSW anomalies and convergence, and to cause larger evaporation and vertical moisture transport.

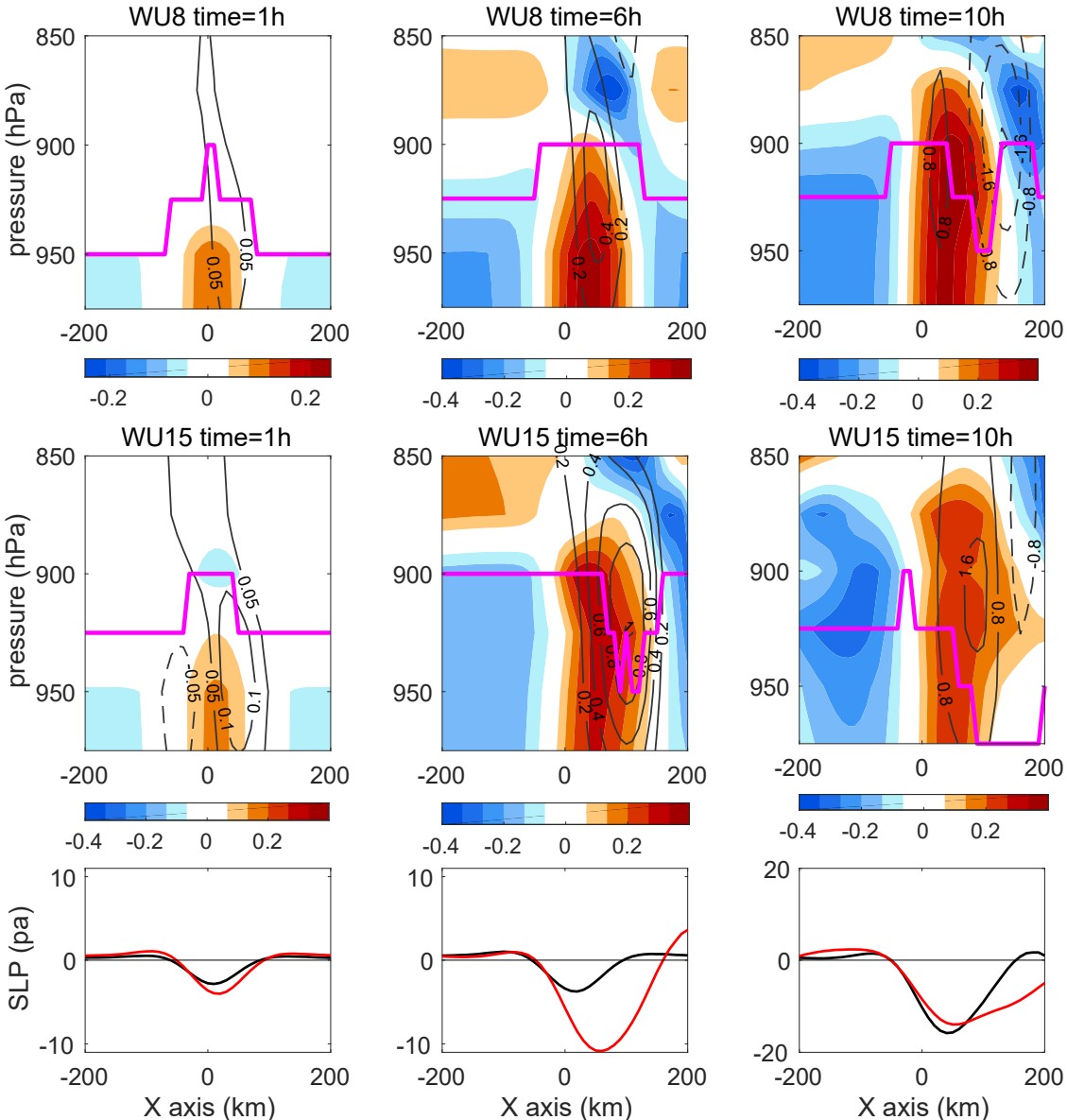

**Figure 10.** Top panel: Time evolution of vertical velocity (W) (contour, unit: $10^{-2}$ m/s$^{-1}$) and PT (color, unit: K) in WU8 along y = 0. Middle panel: Time evolution of vertical velocity W (contour, unit: $10^{-2}$ m/s$^{-1}$) and PT (color, unit: K) in WU15 along y = 0. Bottom panel: SLP anomaly along y = 0 (black for WU8 and red for WU15, unit: pa). Zero is marked by the thin black line. Position of PBLH is marked by thick magenta line in the upper two panels.

### 4.3. Suppressed Vertical Mixing in the Wet Run

In this section, we explore the reason why the vertical mixing in the boundary layer is lower in stronger wind cases. Figure 7 shows a suppressed vertical mixing and lower PBLH and SSW in the wet run than in the dry run. Why is the surface wind anomaly and PBLH smaller with increasing wind in the wet run? To investigate this, we checked the evolution of PBLH over time in WU8 and WU15. In Figure 10, at the first stage of simulation, the PT, vertical motion, SLP, and PBLH are similar to those in the dry run. However, as the integration continued, the PBLH increased not only in the eddy area, but also in the entire simulation region. In the 6th hour of simulation, which is 2 h after the onset of diabatic heating, the PBLH in WU15 began to decrease together with the onset of rain. In the 10th hour of simulation, the PBLH decreased below 950 hPa on the downwind side of the eddy, corresponding

to the downward branch of the SC in WU15 and WU8. This suggests that the decrease in PBLH is influenced by the enhanced vertical motion in the wet run.

How did the vertical motion influence the PBLH? Figure 11 shows the evolution of the zonal wind anomaly (U) and vertical gradient of PT (showing the baroclinic instability in dry and wet runs (DU15 and WU15)). In DU15, U decreased in the upper level and increased at the surface (Figure 11), which is the result of vertical mixing and downward momentum transport. The wind anomalies near the surface increased with time, and the gradient of PT decreased in the boundary layer because of the heating effect of SST. The negative PT gradient decreased stability and reinforced vertical mixing, which in turn increased SSW. In WU15, during the first 4 h of simulation, U and the vertical gradient of PT were similar to those in the dry run. However, after the onset of diabatic heating (after 4 h of simulation), the negative PT gradient began to rise above the PBLH (900 hPa). After the 6th hour, a positive PT gradient appeared under the negative gradient. This is because the diabatic heating in the troposphere increased the PT above the PBLH. Thus, in the wet run, when strong wind triggered diabatic heating above the boundary layer, it caused instability in the troposphere but increased the stability in the boundary layer. This process suppressed the vertical mixing in the boundary layer and reduced the momentum transported down to the surface. This is the reason why the SSW in the strong wind wet runs is lower. This also explains why the Eady growth rate is reduced in these cases.

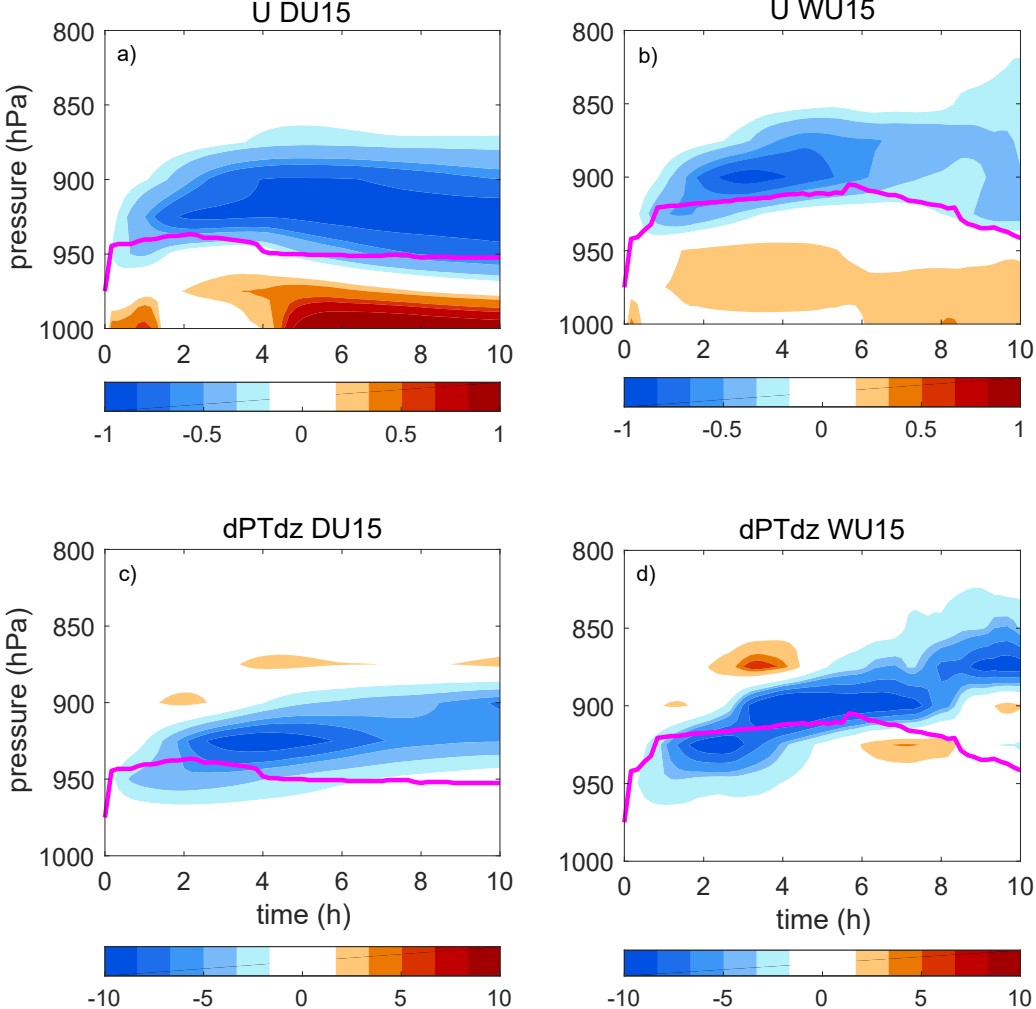

**Figure 11.** Time evolution of U anomaly in DU15 (**a**) and in WU15 (**b**); Time evolution of vertical gradient of PT in DU15 (**c**) and in WU15 (**d**). All these values are averaged in eddy center region (y = 0, x = −200 km~200 km). Unit: m/s for U, and K/m for PT vertical gradient. Time evolution of PBLH (averaged in eddy center region) is marked by the magenta line.

### 5. Conclusions and Discussions

In this paper, we focused on the role of background wind speed and moisture on eddy-induced atmospheric response. Using the WRF model, two parallel sets of idealized experiments were conducted, each including three subexperiments differing only in the strength of the background wind. One set used a dry configuration (dry run), while the other included humidity in the initial condition and considered evaporation to investigate the role of moisture (wet run). The atmosphere in dry and wet runs were forced by the same mesoscale SST anomaly, which resembled the oceanic warm eddy in KE region. Idealized, dry experiments show that under the forcing of very weak wind, the warm eddy can induce a PAM type of atmospheric response, with the SLP anomaly and upward velocity located aloft of the eddy center. Under the forcing of normal or strong winter wind, the warm eddy drives the VMM type of atmospheric response, in which the SLP anomaly and upward velocity shifted to the downwind side of the eddy.

The results of the dry run experiments also suggest an increase in vertical mixing within the boundary layer with increasing background wind speed, which agrees with the findings of Kilpatrich et al. [10]. The increased vertical mixing results in a higher boundary layer height and a larger SSW anomaly over the warm eddy in the stronger wind case. The SSW anomaly induces larger convergence and drives a stronger vertical velocity at the downwind side of the eddy. In turn, this process increases temperature and reduces instability. Thus, the increase in the background wind in dry experiments induces a larger vertical motion. However, because the effect of the background wind is mainly on the vertical mixing in the boundary layer, the vertical motion in the dry run mainly influences the level under 850 hPa.

When the humidity and evaporation were included in the experiments, the pattern of atmospheric responses did not change: The PAM-type responses were still generated in the weak wind case, and the VMM type was generated in stronger wind cases. The difference in atmospheric responses between the dry run and wet run lies in the increase in strength and extent of the vertical motion. In the wet run, the vertical velocity increased approximately 3 times, and the secondary circulation reached up to 800 hPa in normal and strong wind cases. This difference is not only caused by the increased latent heat flux from the surface, but also the diabatic heating in the troposphere triggered by the strong vertical velocity. Only when the background wind is the climatological value or higher, the vertical velocity above the oceanic eddy can be strong enough to induce the water phase change and the precipitation in the troposphere. The diabatic heating in turn reinforces the vertical motion. With this positive feedback, the vertical motion can break through the boundary layer and influence the troposphere. The role of the background wind in the wet run not only induces a strong upward flow and set on precipitation, but also provides upward moisture transport and surface latent heat flux into the atmosphere.

In the wet run, when diabatic heating was triggered above the boundary layer, the heating increased the temperature in the troposphere and caused inversion and increased the stability in the boundary layer. The increased stability suppressed the vertical mixing and reduced the momentum transported downward to the surface. Thus, the strong background wind in the wet run also reduces the surface wind speed.

In this work, we focused on the evolution (the first 10 h) of atmospheric response to oceanic eddies. In the future, we will do more experiments to investigate the eddy's effect on longer time scale atmospheric circulation. Lambaerts et al. [14] examined the sensitive of atmospheric response to vertical wind shear. In their work, when wind shear increased, the correlation between vertical velocity in the PBL and the SST Laplacian decreased more rapidly. In our future work, we will test the sensitivity of wind shear, both in dry and wet conditions.

**Author Contributions:** Conceptualization, Y.J. and Q.L.; data curation, Y.J., L.C., and X.Y.; formal analysis, Y.J.; investigation, Y.J. and L.C.; methodology, Y.J., L.C., and Q.L.; software, L.C.; supervision, Q.L.; writing—original draft, Y.J. and Y.W.

**Funding:** This research was funded by the National Key R&D Program of China (2017YFC1404101), and the National Natural Science Foundation of China (41975065, 41490643, 41176004).

**Acknowledgments:** Thanks to NCAR/UCAR RDA for providing the CFSR reanalysis at rda.ucar.edu/datasets/ds093.1. Thanks to Springer Nature Author Services for language editing. The authors are grateful to the two anonymous reviewers who provided insightful and constructive comments that helped to improve this manuscript.

**Conflicts of Interest:** The authors declare no conflicts of interest.

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
