# Peer review of "The Role of Background Wind and Moisture in the Atmospheric Response to Oceanic Eddies During Winter in the Kuroshio Extension Region"

_atmosphere, doi:10.3390/atmos10090527_

Round 1

Reviewer 1 Report

This manuscript documents how background wind and moisture modulate the atmospheric responses to ocean eddies by conducting sensitivity experiments of idealized WRF simulations. Despite it is written in good logic, a few comments and suggestions are listed below for further revision:

Title:

Instead of using “background”, “ambient” could be more precise.

Abstract:

Line 22: Suggests to replace “vertical flow” with “vertical motion”.

Introduction:

Line 75-76: Please rewrite “in previous modeling work about atmospheric responses over eddies, none  have examined the difference between atmospheric responses under dry and wet conditions. ” Line 85: “background sounding” can be alternated by “ambient atmosphere environment” Line 92: Please include major parameterization schemes that are selected in the current study

Model configuration:

Line 94: “65 sigma levels spans between surface to 10 km height” Line 100: “with evaporation and background humidity profiles”can be simplified as “with inclusion of moisture” Line 105: on “each given” pressure level “vertical wind shear” can be another sensitive parameter to be examined. Have you ever done related study? Or any reference you could think of? Symbols in equations 2 to 5 are not consistent with what paragraphs say, please correct it. Please remove units in subtitle of each plot in Figure 1 since it is already marked at bottom.

Effect of background wind:

Figure 4: 1) The labels look blur with bold style, please modify it. 2) Also, add a sentence to address what magenta line shows in the caption. 3) Why the x-axis in the third row is not identical with other two? Figure 5: Same issue of blur labels. Line 218: Please address the order of N-1 and vertical shear of u. Line 233: Could be modified as “To induce upward motion in the free troposphere, additional heating above PBL is necessary.” Figure 6: 1) Can you explain why an area of large N-1 exists without colocation with vertical motion near bottom-left corner in subplot for DU15? 2) In caption: Vertical profile of N-1 (color “shading”, unit: s-1)

Effects of moisture and diabatic heating

Figure 8: One side of frame in middle-top subplot is missing. Also, labels look blur as well. Figure 8: It’s useful to plot rain/cloud to explain why PBLH is much higher in downwind side in case of WU15 Line 280, 281: w’ (perturbed vertical velocity) and q’ (perturbed water vapor mixing ratio) Figure 9: 1) Why calculating averaged rain rate with longer distance than the others? 2) please use consistent pressure range for the vertical axis with Figures 6 and 8. Line 289: Please modify: time evolution of “diabatic heating (H)” Line 384: Precipitation has been mentioned few times in the paper, it is necessary to include result of simulated cloud/rain.

Language/typos:

Line 59: work “done” by.. Line 73: “can rise to” could be replaced by “can be extended to” Line 207: advection effect “becomes” more important Line 222: vertical “motion” Line 267: makes the SC in the wet run “look more distinct” Line 285: and increase with time “as it extends to level above 800 hpa” Line 288: Please remove “f” in between vertical and velocity Line 299: However, it increased and increased Line 379: “climatological” Line 397: Please revise “Thanks are due to…”

Reviewer 2 Report

This is a first review of “The role of background wind and moisture in the atmospheric response to oceanic eddies during winter in the Kuroshio Extension region” by Jia et al.

This study investigated the role of background wind and moisture in the atmospheric response to oceanic eddies in the Kuroshio Extension region, by performing numerical experiments. Authors’ experiments design is reasonable and their results provided a good perspective on the understanding of the air-sea coupled system. I am a little concerned about their manuscript. I think this paper should be published in Atmosphere after proper modifications.

Major concerns.

1) Authors focused on evolution on a short time scale (~10 hours). Why? In other words, if authors are interested in the development of cyclones passing over the ocean eddy, it is acceptable reasonably. However, it is not. Authors should discuss differences between steady responses after enough model spin-up, e.g., after two weeks later?

In section 4, the authors described the influence of rain. But it is rain reflecting the initial condition, and it is not “pure” rain that fell in the model. Authors should investigate how moisture generated in your model fall as rain? If not, authors need to show that the results are not sensitive to the initial conditions.

2) As you know, there are many moistures over the ocean. Under such situation, why did authors investigate extreme condition, i.e., dry condition? I understand that authors performed extreme experiments (dry experiment) to highlight/extract the influence of moistures. Are there any other specific reasons?

Minor comments

Line 54: Kuroshio -> KE?

Section 2: Please add descriptions on planetary boundary layer scheme, convective parameterization scheme, and other schemes.

Caption in Fig. 2: Please add descriptions on an altitude of vertical velocity.

Caption in Fig. 2: Line 156 : (a)-(c) -> (a)-(d)?

Line 163: larger -> stronger?

Caption in Fig. 3: Line 170: (a)-(c) -> (a)-(d)?

Line 177: strongest-> largest?

Caption in Fig .4. No description about the magenta line.

Line 206: I sense a slight discomfort at this “only”. Regardless of the strength of the wind, SST anomaly, i.e., ocean temperature, influences the overlying atmosphere via a heat exchange between ocean and atmosphere and SST front. Paradoxically speaking, can you reproduce the atmosphere pattern without SST anomaly, anomaly = 0.

Figure 5: How did you calculate the time derivative of “5hr”? Is that a difference between the atmosphere in 5 hr and in 0? Please add information on ⊿t.

Line 220-221: Please indicate ⊿z that you have adopted in equation (7). Is that a difference between 1000hPa and 700 hPa surfaces?

Line 229: Where are you talking about? Over eddy?

Caption in Fig. 7: Line 260: (a)-(c) -> (a)-(d)?

Caption in Fig. 9: Line 289: H_Diabatic heating -> Diabatic Heating (H)?

Fig. 11: Please superimpose the PBLH. That helps readers understand authors’ conclusions.
